# Intimate Partner Violence (IPV) in Military and Veteran Populations: A Systematic Review of Population-Based Surveys and Population Screening Studies

**DOI:** 10.3390/ijerph19148853

**Published:** 2022-07-21

**Authors:** Sean Cowlishaw, Isabella Freijah, Dzenana Kartal, Alyssa Sbisa, Ashlee Mulligan, MaryAnn Notarianni, Anne-Laure Couineau, David Forbes, Meaghan O’Donnell, Andrea Phelps, Katherine M. Iverson, Alexandra Heber, Carol O’Dwyer, Patrick Smith, Fardous Hosseiny

**Affiliations:** 1Phoenix Australia—Centre for Posttraumatic Mental Health, Department of Psychiatry, The University of Melbourne, Level 3, Alan Gilbert Building, 161 Barry Street, Carlton, VIC 3053, Australia; issi.rose@gmail.com (I.F.); dkartal@unimelb.edu.au (D.K.); alyssa.sbisa@unimelb.edu.au (A.S.); annelc@unimelb.edu.au (A.-L.C.); dforbes@unimelb.edu.au (D.F.); mod@unimelb.edu.au (M.O.); ajphelps@unimelb.edu.au (A.P.); carol.odwyer@unimelb.edu.au (C.O.); 2Atlas Institute for Veterans and Families, Royal Ottawa Mental Health Centre, 1145 Carling Avenue, Ottawa, ON K1Z 7K4, Canada; ashlee.mulligan@theroyal.ca (A.M.); maryann.notarianni@theroyal.ca (M.N.); patrick.smith@theroyal.ca (P.S.); fardous.hosseiny@theroyal.ca (F.H.); 3Women’s Health Sciences Division, National Center for PTSD, Veterans Affairs Boston Healthcare System, 150 South Huntington Street, Boston, MA 02130, USA; katherine.iverson@va.gov; 4Department of Psychiatry, Boston University School of Medicine, 72 E Concord Street, Boston, MA 02118, USA; 5Department of Psychiatry and Behavioural Neurosciences, McMaster University, Hamilton, ON L8N 3K7, Canada; alexandra.heber@uregina.ca; 6Veterans Affairs Canada, Charlottetown, PE C1A 8M9, Canada

**Keywords:** intimate partner violence, veteran, military, prevalence, coercive control

## Abstract

Intimate partner violence (IPV) may be a major concern in military and veteran populations, and the aims of this systematic review were to (1) provide best available estimates of overall prevalence based on studies that are most representative of relevant populations, and (2) contextualise these via examination of IPV types, impacts, and context. An electronic search of PsycINFO, CINHAL, PubMed, and the Cochrane Library databases identified studies utilising population-based designs or population screening strategies to estimate prevalence of IPV perpetration or victimisation reported by active duty (AD) military personnel or veterans. Random effects meta-analyses were used for quantitative analyses and were supplemented by narrative syntheses of heterogeneous data. Thirty-one studies involving 172,790 participants were included in meta-analyses. These indicated around 13% of all AD personnel and veterans reported any recent IPV perpetration, and around 21% reported any recent victimisation. There were higher rates of IPV perpetration in studies of veterans and health service settings, but no discernible differences were found according to gender, era of service, or country of origin. Psychological IPV was the most common form identified, while there were few studies of IPV impacts, or coercive and controlling behaviours. The findings demonstrate that IPV perpetration and victimisation occur commonly among AD personnel and veterans and highlight a strong need for responses across military and veteran-specific settings. However, there are gaps in understanding of impacts and context for IPV, including coercive and controlling behaviours, which are priority considerations for future research and policy.

## 1. Introduction

Intimate Partner Violence (IPV) can reference any behaviour that occurs in a current or former intimate relationship that causes physical, psychological, or sexual harm [1], and subsumes acts of physical aggression, sexual coercion, and diverse forms of psychological or emotional abuse. The latter may encompass degrading, humiliating, and threatening behaviours, as well as coercive and controlling behaviours that are intended to dominate the victim and restrict their autonomy [2]; for example, by isolating a person from family and friends, monitoring movements, and restricting access to finances or critical resources. Exposure to IPV is a cause of injuries and physical conditions, and also contributes to mental health issues including depression, posttraumatic stress disorder (PTSD) [3,4], and suicidality [5]. Although research has traditionally addressed consequences of physical violence for victims, there is evidence that psychological abuse and coercive behaviours can have additional impacts on mental health [6,7]. Such impacts contribute towards the significant economic consequences of IPV, which are partly attributed to direct costs from demands on medical and justice systems, and many indirect costs; for example, via workplace impacts and reduced productivity [8,9].

Accumulating evidence indicates that IPV may be a significant concern among active duty (AD) military personnel or veterans, including violence perpetration and IPV victimisation experiences [10,11]. Much of this originates from the U.S. and suggests that IPV may be elevated in AD personnel or veterans, relative to those who never served in the military [11], while IPV perpetration (although not necessarily victimisation) may be elevated among veterans, related to AD personnel [12,13]. However, there remains uncertainty about the extent of these issues across populations, with recent reviews indicating variable findings from studies of different types of IPV and samples including men and women. By way of example, a review by Sparrow et al. [12] considered research regarding IPV victimisation among AD personnel and veterans and reported varying estimates of prevalence across studies of physical types of IPV, with past-year rates ranging from 8% to 39% among women (19% to 38% among men). Similar variability was observed for other types of violence, with past-year rates of psychological IPV victimisation ranging from 9% to 86% among women (6% to 25% among men). The authors quantitatively synthesized data from studies of physical IPV victimisation only and produced mean estimates of 16.2% for women (21.0% for men) when pooling data across studies. A subsequent review of studies of IPV perpetration also identified variable rates of physical violence [13], with relevant figures ranging from 5% to 58% across studies, and mean estimates of 22% and 27% for women and men, respectively. Varying estimates of sexual IPV (12.1% to 40.2% across studies) and psychological IPV perpetration (66.4% to 89.5% across studies) were also reported, although meta-analyses of these data were not conducted. This is notwithstanding that such figures are critical to demonstrate the likely overall extent of the problem across settings and inform policy regarding the appropriate scale and targeting of responses in military and ex-service contexts.

Variations in sampling design may contribute to heterogeneous estimates of IPV victimisation and perpetration among AD personnel and veterans, with convenience sampling likely to inflate estimates in the aforementioned reviews. In contrast, population-based sampling approaches, which clearly define a target population and sampling frame, and use random or systematic sampling strategies, are considered the gold standard in generating findings that are representative of the target population [14]. Methodological approaches that can generate representative figures may also include studies that involve systematic screening of a population, such as veterans seeking services in specific health settings. In the U.S., for example, population screening for IPV among women has been widely implemented in Veteran’s Health Administration (VHA) services [15], and the resulting data provides evidence regarding IPV disclosures among services users. The latter comprise an important sub-population of AD personnel or veterans (i.e., that attend health services and are willing to disclose IPV), with resulting findings that have implications for interventions in health service contexts [16].

Heterogeneous findings may also result from other methodological features of prevalence surveys, which can include varying approaches to measurement of IPV. These have evolved given concern about interpretation of population surveys in non-military settings [17], which often identify comparable rates of IPV across women and men [18]. Such findings of ‘gender symmetry’ in IPV are inconsistent with so-called agency data (e.g., shelter or police records), which routinely indicate that IPV is a heavily gendered dynamic perpetrated primarily by men against women [19]. These discrepancies have drawn attention to features of sampling frames and measures that have been used in many population surveys; for example, that commonly fail to address ‘impacts’ of violence (e.g., fear and injury), which are typically greater among women [20]. Many survey measures have also been criticised for failing to distinguish the context of IPV, and specific forms that reflect ongoing patterns of coercive and controlling behaviour [20]. The latter contexts for IPV are often described in terms of intimate terrorism [21], or coercive control [2], are most likely to come to attention of authorities, and are the main form of IPV encountered in specialist services or justice settings [22,23]. In contrast, coercive controlling violence may characterise a smaller proportion of cases in population samples, which also identify ‘situational’ violence that does not necessarily occur in the context of motivations to assert power and control [24]. Situational violence (e.g., that reflects poor conflict resolution strategies) is often used by both individuals within a relationship, and thus may be common among women and men [25].

There have been recent studies adopting more advanced approaches to survey design that better capture the complexity of IPV. Among other things, such studies have adopted survey measures that directly address the impacts of IPV (e.g., fear, injuries) [20], as well as controlling behaviours [26]. This research has suggested that while all types of violence are linked with adverse outcomes for victims, the coercive and controlling forms of IPV are characterised by greater mental health and psychosocial consequences [27]. Furthermore, this research has supported assertions that coercive and controlling IPV is less prevalent than situational violence in general population samples but is most frequently perpetrated by men against women [28]. Recent reviews of the prevalence of IPV among AD personnel and veterans have not attended substantively to these nuanced features of measurement, which is notwithstanding the importance of recognising heterogeneous forms of IPV, as well as the unique experiences of women and men.

The main aim of this review is to provide the best available estimates of overall prevalence of recent IPV among AD personnel and veterans (with ‘recent’ defined in terms of past year, past six months, past month, or within the current relationship), when considered across settings, including all international jurisdictions, and studies that are most likely to generate representative findings. A further aim is to ensure that such estimates are contextualised by examination of the different types of IPV, including violence with impact and coercive and controlling behaviours. Specific objectives were to:
(1)Provide pooled estimates of overall prevalence of any recent IPV perpetration and victimisation, respectively, among AD personnel and veterans, based on all available studies from across international jurisdictions, using population-based designs or population screening strategies;(2)Examine variability in findings across different types of IPV, including physical, sexual, and psychological abuse, as well as potential indicators of violence impacts and context; and(3)Explore the distribution of any IPV across sample and study characteristics, including gender, serving status (AD personnel versus veterans), study setting (health services, general military/community-based), and country of origin.


## 2. Materials and Methods

This systematic review adhered to the Preferred Reporting Items for Systematic Reviews and Meta-Analyses (PRISMA) [29] statement, which is an extensively used framework ensuring consistency and rigour in the conduct and reporting of systematic reviews. It was also registered prospectively with PROSPERO (ID registration: CRD42020199214).

### 2.1. Search Strategy

A systematic search strategy was developed to identify all available studies from across jurisdictions that reported estimates of IPV perpetration or victimisation in samples of AD personnel or veterans, and utilised population-based sampling techniques or population screening. Searches of electronic databases were conducted: PsycINFO, CINHAL, PubMed (including MEDLINE), and the Cochrane Library. See Appendix A for search terms. Supplementary searches of reference lists of recent reviews were also conducted [12,13]. Searches were restricted to peer-reviewed journals from inception to 26 May 2020.

### 2.2. Eligibility Criteria

Eligible studies addressed populations comprising AD personnel and/or veterans, and utilised (1) population-based sampling designs, or (2) population screening strategies. For current purposes, population-based sampling designs were defined by probability-based sampling strategies (i.e., described as random, stratified, or systematic sampling) [14] that involved recruitment from either (1a) the general population of AD personnel and veterans, or (1b) AD personnel and veterans recruited from military or veteran-specific health services. Population screening was defined as the systematic questioning of an entire target population, which generally involved patient records derived from screening for IPV among consecutive health service users. Studies that utilised any type of self-report measure of IPV were eligible.

Studies were excluded if they addressed IPV reported exclusively by family members (e.g., veteran spouses), or addressed violence directed towards family members other than partners. This was intended to limit the heterogeneity of studies, owing to the already broad focus on IPV victimisation and perpetration, as well as different types of IPV. Studies were also excluded if they combined multiple forms of family violence (e.g., child exposure, family members other than intimate partner), and did not report IPV separately, or utilised alternative sampling techniques (e.g., convenience sampling). The sampling bias commonly manifested in small studies [30] was addressed via exclusion of samples comprising <100 participants. The representativeness of data derived from population screening studies may be reduced given low response rates, and these studies were also excluded if <60% of eligible population members did not participate [14]. Although response rates were not used to exclude studies involving probability-based sampling, low response rates were considered in risk-of-bias assessments (see Appendix A).

### 2.3. Selection Process

Following a pilot test of eligibility criteria, records were screened on title and abstract. The pilot phase involved around 20% of records being independently screened by two reviewers (I.F. and A.S.), and this identified disagreements in <5% of cases. These disagreements were resolved through discussion, including with a third reviewer where necessary, and with a view towards enhancing consistency in the classification of the remaining records (which were each screened by one reviewer). The same reviewers then independently reviewed full text records for potentially eligible studies using the Covidence [31] tool. All records that could not be excluded on title and abstract were subject to full text review. Any disagreements at full-text screening stages were also resolved by discussion, or through adjudication with a third reviewer. Records deemed ineligible at full-text screening were excluded with the reason recorded.

### 2.4. Risk of Bias Assessment

Eligible studies were assessed for risk of bias by two reviewers (I.F. and D.K.) using the Joanna Briggs Institute’s Critical Appraisal Checklist for Studies Reporting Prevalence Data (see Appendix A for full criteria) [32,33]. The risk-of-bias rating for each criterion was categorised (Yes = 1, No = 0, Unclear = 0) [34] and total scores were utilised to generate an overall risk-of-bias rating for each study (i.e., score ≥ 6 = low risk; <6 = high risk).

### 2.5. Data Extraction

Where available, data were extracted from published reports regarding sample gender (men only, women only, combined, other gender identities), IPV reference period, and type of IPV (e.g., any IPV, physical, sexual, or psychological). Studies that reported IPV prevalence within the past year, past six months, past month, or within the current relationship were all classified as referencing recent IPV. This represents a different approach to defining ‘recent’ violence, when compared to reviews that have considered reference periods including past year [12] and past three years [35]. This reflects an intention to minimise bias associated with omitting informative studies that have measured IPV across slightly different periods. It also aligns with the focus on coercive and controlling forms of IPV, which often reflect enduring patterns of behaviour (whereby violence in the current relationship is likely to reflect ongoing exposure) [2]. All other time points were classified as lifetime IPV, which are included among the eligible studies, but were not incorporated in the analyses. When necessary, study authors were contacted to request relevant data. Data were also extracted regarding sample size, country of origin, service status (AD, veteran), era of service, study setting (health service, general military/community-based), and sampling strategy. Further data were also extracted regarding IPV measures, including any information reported regarding IPV impact and context for violence. Potential indicators of the latter included measures of controlling behaviours, and reports of fear of partner that can distinguish coercive and controlling IPV [36]. One reviewer (I.F.) extracted data from each study.

### 2.6. Evidence Syntheses

Random effects meta-analyses were used to quantitatively synthesize estimates of recent IPV prevalence across comparable studies, and produce weighted mean estimates, with corresponding 95% Confidence Intervals (CI), using the *metaprop* command in Program R [37]. True heterogeneity (not attributable to sample error) in study-specific estimates was quantified using the I^2^ statistic [38]. I^2^ values of 25%, 50% and 75% may represent low, moderate, and high levels of heterogeneity.

The first stage of analyses was focused on quantitatively synthesizing estimates of any IPV perpetration and victimisation, respectively. Data were categorized this way if authors referred expressly to ‘any IPV’. Alternatively, when findings for multiple types of IPV were reported (e.g., physical, psychological, sexual), the highest figure was used as the best available estimate. Where population-based studies reported findings adjusted by survey weights, the unweighted sample size and number of cases were used to calculate prevalence, given that *metaprop* requires data comprising number of events and sample size, which are not suitable for synthesis subsequent to application of sampling weights. In instances where minimal data required for meta-analyses were not reported, and authors could not be contacted, attempts were made to calculate figures from other published data. Pooled estimates and 95% CIs were calculated initially across all studies with relevant data (men, women, and combined genders), while additional analyses were produced for studies that reported data for men only and women only.

In subsequent analyses, a series of quantitative syntheses which were disaggregated by gender were conducted to produce estimates of specific IPV subtypes (physical, sexual, psychological IPV). Findings regarding IPV impact and context were synthesised narratively given the heterogeneity of measures. A series of planned subgroup analyses were then conducted to examine variability in any IPV according to moderators. These included: (1) gender (men only vs. women only); (2) era of service (pre-2001 vs. post-2001); (3) military status (AD personnel vs. veterans); (4) study setting (military/veteran health service vs. general military/community-based); and (5) country of origin (U.S. vs. others). Finally, sensitivity analyses were conducted to explore whether the main quantitative findings were robust to risk of bias and outliers. The latter were defined as studies with 95% CIs that did not overlap with the 95% CI for the pooled estimate [39].

## 3. Results

### 3.1. Search Results

Figure 1 displays the PRISMA flowchart of search results. As shown, the searches yielded 2168 records, and 1585 (minus duplicates) were screened on title and abstract. There were 223 records that could not be excluded on this basis and were subject to full-text review. This identified *k* = 32 studies eligible for inclusion. Data from *k* = 31 studies, involving *n* = 172,790 participants, were sufficiently comparable and available for quantitative syntheses. This included findings from two unique datasets reported in a single paper [40]. Findings from two studies that did not report data to support inclusion in the meta-analyses are also summarized in Table 1 [41,42]. Heterogenous data from *k* = 11 studies, involving *n* = 132,381 participants, were available regarding the impact and context for violence, including coercive and controlling behaviours.

### 3.2. Study Characteristics

Table 1 provides an overview of individual study characteristics, while frequency analyses of study characteristics are provided in Appendix A.

#### 3.2.1. Sample Characteristics

Most studies (*k* = 26, 83.9%) originated from the U.S., with smaller numbers from Canada (*k* = 3, 9.7%), the UK (*k* = 1, 3.2%), and Turkey (*k* = 1, 3.2%). Studies were conducted in military or veteran-specific health services (*k* = 10, 32.3%), and general military settings (e.g., personnel sampled from military installations) or community contexts (e.g., community surveys identifying individuals with a history of service, *k* = 21, 67.7%). More than one third of studies (*k* = 12, 38.7%) were restricted to samples of AD personnel or veterans that self-identified as married or in an intimate relationship.

#### 3.2.2. IPV Perpetration

Twelve studies assessed IPV perpetration. Although there were some mixed gender studies addressing perpetration, there were five times more men (*n* = 112,187) than women (*n* = 19,588). There were no studies reporting data regarding other gender identities. Physical IPV was the most assessed subtype (*k* = 10; 83.3%). Perpetration was commonly assessed using the Conflicts Tactics Scale (CTS; *k* = 5, 41.7%), while one third of studies (*k* = 4, 33.3%) employed brief screening measures. Additional studies employed the Family Maltreatment Measure (*k* = 2, 16.7%) and Abusive Behaviour Inventory (*k* = 1, 8.3%). Example items from brief measures included: “Have you ever perpetrated violence against your wife/girlfriend?” [43]; “When arguing, do you yell, hit objects, or throw/break things?”; “Has your partner ever been afraid of your anger in the past year?”; and “Have you pushed, grabbed, slapped, or punched your partner in the past year?” [44].

There were six studies (*n* = 99,499) that reported findings regarding impact and context for violence perpetration, and details of these are in Appendix A. This included one study that reported impacts of IPV in terms of causing injury, as measured by the injury subscale of the CTS-2 [45]. There were two studies that measured ‘clinically significant’ IPV, which was defined by physical violence with impact. The earliest used questions that were similar to the Physical Assault subscale of the CTS-2 and were followed by items addressing resulting injuries [46]. Clinically significant IPV was defined by physically violent acts that were associated with high inherent dangerousness (e.g., use of a weapon), or physical injury. The second study used a validated scale which comprised similar items addressing physically violent behaviours and impacts [40]. For purposes of this measure, clinically significant IPV was defined by physical violence that resulted in injury, significant fear, or had high potential for injury (e.g., burning, choking, use of a weapon). While there were no studies that reported findings from comprehensive measures of coercive and controlling behaviours, there was one study that included survey items addressing whether they demanded to know ‘who and where’ their partner was at all times [47], while another study reported findings regarding whether the partner had been afraid of their anger [44].

Given the diversity of indicators of IPV impact and context, these data were not included in meta-analyses and are instead summarised narratively. However, the aforementioned studies also reported findings regarding prevalence of any IPV, or subtypes of IPV, which were included in the meta-analyses. The one exception was for figures from one study [40] regarding clinically significant IPV. This was the only available estimate of IPV prevalence that was reported and was thus included in the meta-analyses.

**Table 1 ijerph-19-08853-t001:** Characteristics of eligible studies.

Author Country	Study Description	*n* (% Men)	IPV Assessment	IPV Prevalence		Overall RoB
**Studies reporting IPV Perpetration and Victimisation (*k* = 6)**						
				**Perpetration**	**Victimisation**	
Creech [45] *U.S.*	Random sample of veterans (Army, Navy, Air Force, Coast Guard) in an intimate relationship	*n* = 102 (0%)	CTS-2 IPV perpetration and victimisation in past 6 months Current partner	Any IPV: 71.8% Physical IPV: 11.9% Sexual IPV: 11.9% Psychological IPV: 67.9% Caused injury: 2.0%	Any IPV: 66.9% Physical IPV: 9.9% Sexual IPV: 16.5% Psychological IPV: 64.0% Experienced injury: 2.0%	5/9
Foran [46] *U.S.*	Representative sample of AD Air Force personnel in an intimate relationship (married, engaged or intimate partner)	*n* = 42,744 (81.2%)	Modified CTS Past-year IPV perpetration and victimisation Current partner	**Men** Physical IPV: 12.9% * Severe physical IPV: 4.7% * **Women** Physical IPV: 15.1% * CS physical IPV: 3.3% *	**Men** Physical IPV: 19.6% * Severe physical IPV: 3.5% * Emotional IPV: 6% * **Women** Physical IPV: 18.3% * CS physical IPV: 3.5% * CS emotional IPV: 8.5% *	8/9
Gerlock [48] ^a^ *U.S.*	Random sample of veterans (Army, Marine, Navy, Air Force, Coast Guard) in an intimate relationship and in treatment for PTSD	*n* = 441 (100%)	Abusive Behaviour Inventory Past-year & lifetime IPV perpetration and victimisation Current or ex-partner	**Past-year (current partner)** Physical IPV = 27% **Lifetime (current partner)** Physical IPV = 41% **Lifetime (ex-partner)** Physical IPV = 43%	**Lifetime** Physical IPV = 36%	4/9
	Spouses of a random sample of veterans (Army, Marine, Navy, Air Force, Coast Guard) in treatment for PTSD	*n* = 441 (0%)	Abusive Behaviour Inventory Past-year & lifetime IPV perpetration and victimisation Current partner	**Lifetime** Physical IPV = 34%	**Past-year** Physical IPV = 27% **Lifetime** Physical IPV = 47%	
Lorber [40] *U.S.*	Random sample of AD Air Force personnel with intimate partners (married or living with intimate partner) and children (data collected in 2008)	*n* = 25,285 (81.8%)	Family Maltreatment Measure Past-year IPV perpetration and victimisation Current partner	**Combined gender** Physical IPV: 1.1% *	**Combined gender** Physical IPV: 2.0% * Emotional IPV: 7.2% *	7/9
Lorber [40] *U.S.*	As above (data collected in 2011)	*n* = 29,359 (84.6%)	Same as above	**Combined gender** Physical IPV: 0.5% *	**Combined gender** Physical IPV: 1.4% * Emotional IPV: 7.0% *	7/9
Zamorski [47] *Canada*	Population-based survey of a random sample of Canadian AD personnel in an intimate relationship	*n* = 1745 (87.8%)	Modified CTS Current relationship IPV perpetration and victimisation Current partner	**Men** Physical or sexual: 9.5% * Emotional or financial: 19.4% * **Women** Physical or sexual: 9% * Emotional or financial: 18.8% *	**Men** Physical or sexual: 16.4% * Emotional or financial: 25.6% * **Women** Physical or sexual: 7.5% * Emotional or financial: 22.0% *	8/9
**Studies reporting IPV perpetration only (*k* = 6)**						
				**Perpetration**		
Cancio [49] *U.S.*	Nationally representative community sample with a history of involvement in the Armed Forces (veterans)	*n* = 499 (100%)	Single items Past-year IPV perpetration Current partner	Physical IPV: 8.4% Sexual IPV: 4.7%		3/9
Hundt [44] *U.S.*	Routine clinical assessment of Veterans referred to an outpatient mental health clinic	*n* = 264 (91%)	Single items Past-year IPV perpetration Current partner	**Combined gender** Any IPV: 42% Physical IPV: 17% Partner afraid of veteran’s anger: 42% Yelling or hitting/throwing/breaking objects: 55%		4/9
McCarroll [50] *U.S.*	Representative sample of married AD personnel who had, or had not, deployed	*n* = 1025 (100%)	CTS Domestic violence perpetration pre-deployment (lifetime) and post-deployment (past month) Current partner	Pre-deployment lifetime physical IPV: 10.6% Post-deployment past month physical IPV: 7.2%		6/9
McCarroll [51] *U.S.*	Random sample of married AD personnel sampled from Army installations	*n* = 26,835 (95.1%)	CTS Past-year IPV perpetration Current partner	**Men** Mild physical IPV: 18.3% Severe physical IPV: 5.2% **Women** Mild physical IPV: 24.2% Severe physical IPV: 8.0%		6/9
Ortabag [43] ^b^ *Turkey*	Representative sample of military personnel at a Turkish Military Medical Academy	*n* = 637 (100%)	Single items Lifetime IPV perpetration Current partner	Any IPV: 8.8%		5/9
Schmaling [52] *U.S.*	Sample of reservist military personnel in an intimate relationship (married or living with an intimate partner) mobilised for deployment	*n* = 2841 (90.6%)	CTS Past-year IPV perpetration Current partner	**Men** Physical IPV: 15.3% **Women** Physical IPV: 20.7%		7/9
**Studies reporting IPV victimisation only (*k* = 19)**						
				**Victimisation**		
Albright [53] *U.S.*	National community sample of randomly selected college students with a history of involvement in the Armed Forces (AD personnel and veterans)	*n* = 2658 (67.1%)	Single items Past-year IPV victimisation Current or ex-partner	**Combined gender** Any IPV: 41.9% Physical IPV: 23.2% Sexual IPV: 8.3% Emotional IPV: 10.7%		3/9
Albright [54] ^a^ *U.S.*	Stratified community sample with a history of involvement in the Armed Forces (veterans)	*n* = 2872 (91.5%)	Single items Lifetime IPV victimisation Current or ex-partner	**Men** Physical or sexual IPV: 8.1% * **Women** Physical or sexual IPV: 32.0% *		7/9
Bartlett [55] *U.S.*	Online research panel of a randomly selected representative community sample with a history of involvement in the Armed Forces, Military Reserves or National Guard (Veterans)	*n* = 642 (100%)	HARK Past-year IPV victimisation Current or ex-partner	Any IPV: 14.9% * Physical IPV: 7.0% * Sexual IPV: 1% * Emotional IPV: 12.0% * Fear of partner: 5.4% *		8/9
Belik [56] ^a^ *Canada*	Representative sample of AD regular and reserve forces members	*n* = 8441 (69.3%)	Single item Lifetime IPV victimisation Current or ex-partner	**Men** Physical IPV: 1.1% * **Women** Physical IPV: 7.1% *		6/9
Bostock [42] ^ϕ^ *U.S.*	Random sample of AD Air Force personnel	*n* = 2018 (0%)	Single items Recent IPV victimisation Husband or boyfriend	Sexual IPV (rape): 7.3%		N/A
Campbell [57] ^a^ *U.S.*	Random sample of AD personnel (Air Force, Army, Navy, Marines)	*n* = 616 (0%)	Modified Abuse Assessment Screen Lifetime & during military service IPV victimisation Current or ex-partner	**Lifetime** Any IPV: 38.8% Physical IPV: 26.9% Sexual IPV: 12.3% Emotional IPV: 33.1% **IPV during military** Any IPV: 21.6%		6/9
Campbell [58] ^a^ *U.S.*	Random sample of veterans and reservists (Army, Navy, Air Force, Marines) attending a Veteran Affairs clinic	*n* = 268 (0%)	CTS-R Lifetime IPV victimisation Current or ex-partner	Physical IPV: 74%		7/9
Cerulli [59] ^a^ *U.S.*	Stratified community sample with a history of involvement in the Armed Forces (veterans)	*n* = 4356 (100%)	Single items Lifetime IPV victimisation Current or ex-partner	Physical or sexual IPV: 9.5%		6/9
Dichter [60] ^a^ *U.S.*	Stratified community sample of women with a history of involvement in the Armed Forces (veterans)	*n* = 503 (0%)	Single items Lifetime IPV victimisation Current or ex-partner	Physical or sexual IPV: 33%		5/9
Dichter [61] *U.S.*	VHA routine screening of veterans	*n* = 541 (0%)	E-HITS Past-year IPV victimisation Current or ex-partner	Positive screen for IPV: 16.6%		5/9
Dichter [6] *U.S.*	VHA routine screening of veterans	*n* = 8888 (0%)	E-HITS Past-year IPV victimisation Current or ex-partner	Positive screen for IPV: 8.7% Physical IPV: 1.5% Sexual IPV: 1.1% Psychological IPV: 6.2%		7/9
Dighton [62] ^a^ *UK*	Representative community sample with histories of involvement in the Armed Forces (veterans)	*n* = 257 (82.7%)	Single item Lifetime IPV victimisation Current or ex-partner	**Men** Physical IPV: 21.6% * Death threats: 3.2% * Money withheld by partner: 7.9% * **Women** Physical IPV: 26.1% * Death threats: 13.0% * Money withheld by partner: 18.7% *		5/9
Iverson [5] *U.S.*	Random sample of VHA veterans (Army, Navy, Air Force, Marines, Coast Guard) in intimate relationships	*n* = 160 (0%)	CTS Past-year IPV victimisation Current partner	Any IPV: 28.8% Physical IPV: 14.8% Sexual IPV: 14.4% Psychological IPV: 18.1%		7/9
Iverson [63] ^a^ *U.S.*	Nationally representative sample of veterans (Army, Navy, Air Force, Marines, Coast Guard) in GfK KnowledgePanel	*n* = 411 (0%)	HARK Lifetime IPV victimisation Current or ex-partner	Any IPV: 54.7% Physical IPV: 21.2% Sexual IPV: 29.4% Psychological IPV: 47.2% Stalking: 35.3%		8/9
Iverson [64] *U.S.*	Random sample of veterans (Army, Navy, Air Force, Marines) in intimate relationships (married or intimate partner)	*n* = 407 (47.9%)	CTS 6 months IPV victimisation Current partner	**Men** Any IPV: 65.6% Physical IPV: 7.7% Sexual IPV: 4.1% Psychological IPV: 64.6% **Women** Any IPV: 60.0% Physical IPV: 7.1% Sexual IPV: 7.1% Psychological IPV: 58.8%		7/9
Kimerling [65] *U.S.*	National population-based sample of veterans (Army, Navy, Air Force, Marines)	*n* = 6287 (0%)	HARK Past-year IPV victimisation Current or ex-partner	Positive screen for IPV: 18.5% * Physical IPV: 4.9% * Sexual IPV: 2.2% * Psychological IPV: 14.7% * Fear of partner: 9.9% *		8/9
Mercado [66] ^a^ *U.S.*	Random sample of VHA veterans (Army, Navy, Air Force, Marines, Coast Guard)	*n* = 369 (0%)	Single item During military IPV victimisation Current or ex-partner	Sexual IPV (during military): 7.3%		5/9
Rosenfeld [67] *U.S.*	Nationally representative sample of veterans (Army, Navy, Marines, Coast Guard, Air Force) receiving care from VHA	*n* = 2302 (0%)	Single items Past-year reproductive coercion victimisation Any male sexual partner	Reproductive coercion: 11%		5/9
Sadler [68] ^a^ *U.S.*	Random sample of veterans (Army, Navy, Air Force)	*n* = 506 (0%)	Single item IPV victimisation during military service	Sexual IPV (rape): 3.8%		6/9
Sadler [41] ^ϕ^ *U.S.*	Random sample of veterans (Army, Navy, Air Force; 1996–1997)	*n* = 520 (0%)	Single item IPV victimisation during military service	Premilitary domestic violence: 20.6%		N/A
Skomorovsky [69] *Canada*	Random sample of Regular Canadian Armed Forces members in intimate relationships (married or intimate partner)	*n* = 529 (81.9%)	Modified CTS (from General Social Survey) Current relationship IPV victimisation Current partner	**Combined genders** Physical IPV: 13.2% Emotional IPV: 26.2%		4/9

Notes. CTS = Conflict Tactics Scale; CS = clinically significant; E-HITS = Extended–Hurt, Insulted, Threaten, Scream; HARK = Humiliation, Afraid, Rape, Kick; IPV = Intimate Partner Violence; N/A = not applicable; PTSD = Posttraumatic Stress Disorder; RoB = Risk of bias; U.S. = United States; VHA = Veteran Health Administration. ϕ = Study not included in meta-analyses. No quality assessment was conducted. * = Authors reported weighted data and estimates may vary from unweighted figures included in meta-analyses. ^a^ = Study reports lifetime estimates of IPV victimisation and were not included in the meta-analyses. ^b^ = Study reports lifetime estimates of IPV perpetration and were not included in the meta-analyses.

#### 3.2.3. IPV Victimisation

Twenty-five studies assessed IPV victimisation. These included twice as many men (*n* = 97,769) as women (*n* = 42,860) across studies. There was one study [53] that reported data on gender identities other than men or women (transgender: *n* = 31). Victimisation was mostly assessed using un-validated screening measures (*k* = 11, 42.3%), followed by the CTS (*k* = 7, 28.0%), and two brief screening tools that have been validated. These comprised the Humiliation, Afraid, Rape, Kick (HARK) scale (*k* = 3, 12.0%), and the Extended–Hurt, Insulted, Threaten, Scream (E-HITS) scale (*k* = 2, 8.0%). Example items from brief measures (*k* = 9, 36.0%) included: “Has an intimate partner ever threatened you with physical violence?”; “Has an intimate partner ever attempted physical violence against you?”; “Has an intimate partner ever hit, slapped, pushed, kicked, or hurt you in any way?” [54,59,60]. Other measures included the Family Maltreatment Measure (*k* = 2), the Abusive Behaviour Inventory (*k* = 1), and the Abuse Assessment Screen (*k* = 1).

There were *k* = 10 studies (*n* = 108,073) that reported data regarding impacts or context for IPV victimisation (see Appendix A). These included one study that addressed injury from IPV, measured using the injury subscale of the CTS-2 [45]. There were two studies that reported findings regarding clinically significant physical IPV and emotional abuse, as defined by violent behaviours with impact. The earliest defined impacts of physical IPV in terms of injury or inherently dangerous acts such as using a weapon, while impacts of emotional abuse included significant emotional distress [46]. Emotional abuse items were only administered to participants who reported experiencing depression, stress, or fear of their partner’s behaviour. The second study measured clinically significant physical or emotional abuse using the Family Maltreatment Measure [40]. There were two studies that reported findings regarding fear of partner or ex-partner [55,65]. While there were no studies that reported findings from comprehensive measures of coercive or controlling behaviours, there were four studies that measured exposure to specific acts including intimate partner stalking [63], withholding money [62], limiting contact with friends and family, damaging or destroying property, and demanding to know ‘who and where’ they were at all times [47]. These also included one study that measured reproductive coercion via survey items referencing partners withholding or restricting use of birth control [67].

Given the heterogeneity of indicators of IPV impact or context, the specific data were not considered in meta-analyses and were instead summarised narratively. However, the aforementioned studies also reported findings regarding any IPV, or subtypes of violence (e.g., physical IPV), which were included in quantitative analyses. Two exceptions were for figures regarding clinically significant emotional abuse [46], and clinically significant physical or emotional aggression [40]. These were the only data provided regarding prevalence in these studies and were included in the meta-analyses.

#### 3.2.4. Risk of Bias of Included Studies

Appendix A illustrates the risk of bias of included studies (*k* = 31). The majority (*k* = 19, 61.3%) were judged to have low risk of bias, while 12 studies (38.7%) were judged to have high risk of bias. Only two studies were classified as using appropriate statistical analysis (i.e., reported numerator [case number], denominator [sample size], and percentages with 95% CI), while approximately half of studies (*k* = 23) had sufficient coverage of the sample. A minority of studies (*k* = 16) used validated IPV measures.

### 3.3. Evidence Synthesis

A series of random effects meta-analyses were used to generate pooled estimates of the prevalence of recent IPV perpetration and victimisation among AD personnel and veterans. These included summary effects (weighted mean estimates), 95% CIs, and heterogeneity statistics (I^2^). For estimates of any recent IPV, figures are reported initially across all studies with relevant data (men, women, and combined gender), while further analyses have been disaggregated by gender. Analyses disaggregated by gender are also reported for IPV subtypes, while a narrative approach was used to summarise findings regarding impact or context for IPV.

#### 3.3.1. Recent IPV Perpetration

Data from *k* = 11 studies and *n* = 131,140 participants addressed any recent IPV perpetration among AD personnel and veterans, and these reported prevalence estimates ranging from <1.0% to 73% across studies. Meta-analyses produced a mean estimate (weighted by sample size) of 12.7%, with a 95% CI ranging from 4.9% to 29.0%. The I^2^ statistic (100%) indicated high levels of true heterogeneity.

Table 2 reports findings from meta-analyses that were disaggregated by gender and type of IPV. There were *k* = 7 studies that reported data on any recent IPV perpetration among men, and analyses produced a weighted mean prevalence of 15.8% (95% CI = 11.8–20.9%), with high levels of true heterogeneity (I^2^ = 99.4%). There were fewer studies that reported findings regarding physical IPV perpetration (*k* = 6; pooled estimate = 14.3%), with single studies reporting rates of psychological and sexual IPV. Table 2 also shows *k* = 5 studies that reported data on any recent IPV perpetration among women, with a weighted mean prevalence of 28.8% (95% CI = 14.7–48.7%). There were high levels of true heterogeneity (I^2^ = 99.4%). There were fewer studies that reported data among women regarding recent physical IPV perpetration (*k* = 4; pooled estimate = 19.4%), and recent psychological IPV (*k* = 2; pooled estimate = 39.5%), while a single study reported rates of recent sexual IPV perpetration among women.

There were six studies that included measures of impact or context for IPV perpetration and reported findings regarding prevalence (see Appendix A). Foran [46] reported figures from a large survey of U.S. air force personnel in 2006 and identified 4.7% of men and 3.3% of women who were involved in intimate relationships and reported perpetration of clinically significant physical IPV (as defined by physical violence with ‘impact’). Lorber [40] reported findings from subsequent surveys of air force personnel and identified 1.1% and 0.5% that reported clinically significant physical IPV in 2008 and 2011, respectively. Creech [45] identified around 2.0% of women veterans who reported having caused injury to their partner in the past year, based on a small study of U.S. veterans, while Zamorski [47] identified 3.8% of Canadian AD personnel that had demanded to know ‘who and where’ their partner was at all times (over the life of the current relationship). There was one further study that analysed data from routine assessments of veterans conducted during referral to a VHA mental health clinic, and these identified 42% who indicated their partner had been afraid of their anger in the past year [44].

#### 3.3.2. Recent IPV Victimisation

Data from *k* = 14 studies and *n* = 121,649 participants addressed any recent IPV victimisation, and these reported prevalence estimates ranging from 5% to 68% across studies. Meta-analyses produced a mean prevalence estimate (weighted by sample size) of 20.7%, with a 95% CI ranging from 13.1% to 31.1%. The I^2^ statistic (100%) indicated high levels of true heterogeneity across studies.

Table 2 also reports findings from analyses disaggregated by gender and types of IPV. There were *k* = 4 studies that reported data on any recent IPV victimisation among men, and these informed a weighted mean prevalence of 28.1% (95% CI = 13.6–49.3%). There were high levels of true heterogeneity (I^2^ = 99.4%). Four studies reported corresponding figures for recent psychological IPV victimisation among men (pooled estimate = 20.2%), while fewer studies reported findings regarding physical IPV (*k* = 3; pooled estimate = 10.1%), and sexual IPV victimisation among men (*k* = 2; pooled estimate = 0.8%). There were *k* = 9 studies that reported data on any recent IPV victimisation among women, with analyses producing a weighted mean prevalence of 24.2% (95% CI = 14.4–37.7%). There were high levels of true heterogeneity across studies (I^2^ = 99.7%). Smaller numbers of studies reported data regarding recent physical IPV victimisation among women (*k* = 6; pooled estimate = 7.4%), recent psychological IPV (*k* = 7; pooled estimate = 22.0%), and recent sexual IPV (*k* = 5; pooled estimate = 5.2%).

There were *k* = 10 studies that reported prevalence findings regarding impacts or context for IPV victimisation (see Appendix A). Two studies addressed clinically significant physical IPV, which was defined in terms of exposure to violent behaviours with ‘impact’. Foran [46] reported results from a 2006 survey of AD air force personnel who were involved in intimate relationships and identified 3.5% of men and women that reported clinically significant physical IPV victimisation, while 6.0% of men and 8.5% of women reported clinically significant emotional abuse. Lorber [40] reported findings from subsequent surveys of AD air force members who were in married or cohabitating relationships (and had one or more children) and identified 2.0% that reported past year clinically significant physical IPV in 2008 (1.4% in 2011), while 7.2% reported significant emotional abuse in 2008 (7.0% in 2011). There was one small U.S. study of women veterans, and this identified around 2.0% that reported injuries resulting from their partners behaviour [45].

Four studies used IPV scales including items reflecting coercive and controlling behaviours. Zamorski [47] reported findings from a survey of Canadian AD personnel who were in intimate relationships and identified 6.0% of men who reported their partner had limited their contact with friends and family, while 8.2% of men (5.7% of women) reported their partner demanded to know ‘who and where’ they were at all times. There were 4.6% of men (4.1% of women) who indicated their partner had damaged or destroyed their possessions or property. Dighton [62] considered a small sample of UK veterans and identified 7.9% of men and 18.7% of women reported ever having had money withheld by their partner, while Iverson [63] analysed data from a national sample of women veterans and identified 64.4% that reported lifetime experiences of stalking by an intimate partner. Finally, Rosenfeld considered women veterans who had used VHA primary care services and had sex with a man in the past year and identified 11.0% that reported exposure to reproductive coercion (e.g., partner restricted or withheld use of birth control) [67]. Finally, there were two studies that reported findings from questions regarding fear of partner, derived from multi-item scales. Kimerling [65] described a telephone survey of women veterans and identified rates of fear of partner ranging from 2.1% to 14.4% across age groups. Bartlett [55] reported survey findings from male veterans who had been exposed to trauma and identified 5.4% that reported fear of partner in the past year.

### 3.4. Sub-Group Analyses

For purposes of sub-group analyses, data regarding any recent IPV perpetration and victimisation were considered to maximize the number of studies in analyses, which addressed sample gender, military status (AD vs. veteran), study setting (military/veteran health service vs. general military/community-based), and era of service (pre-2001 vs. post-2001). Results from studies that could not be classified into these sub-groups (e.g., mixed-gender samples) are not reported. Findings including pooled estimates and 95% CIs for sub-groups of studies, along with results of conventional significance tests are presented to highlight potentially meaningful differences.

Table 3 indicates a significant difference in rates of IPV perpetration according to serving status and study setting. Higher rates of recent IPV perpetration were reported in studies of veterans (pooled estimate = 31.8%) relative to AD personnel (pooled estimate = 5.2%), *Q* = 7.1, *df* = 1, *p* < 0.05. Higher rates of recent IPV perpetration were also reported in studies of health services (pooled estimate = 33.7%), compared to general military or community contexts (pooled estimate = 9.9%), *Q* = 5.5, *df* = 1, *p* < 0.05. There were no other significant differences regarding rates of any recent IPV perpetration or victimisation. This was notwithstanding trends that may suggest modest differences that were not statistically significant in the context of fewer studies. These included a trend towards higher rates of IPV perpetration among women (pooled estimate = 28.8%) than men (pooled estimate = 15.8%), and towards higher levels of recent IPV victimisation in studies of veterans (pooled estimate = 24.3%), compared to AD personnel (pooled estimate = 13.3%). Finally, there was a trend towards higher rates of IPV victimisation in general military or community contexts (pooled estimate = 24.2%), relative to disclosures in health services (pooled estimate = 15.3%). In these instances, the 95% CIs for sub-group estimates were overlapping, and thus were not statistically significant.

### 3.5. Sensitivity Analyses

Sensitivity analyses examined whether findings were robust to risk of bias and outliers, and also considered data regarding any IPV. These indicated that studies characterized by high risk of bias typically reported higher rates of IPV perpetration (pooled estimate = 37.0%) when compared to studies judged to have low risk of bias (pooled estimate = 6.2%) (*Q* = 8.03, *df* = 1, *p* < 0.005; See Appendix A). There was no significant difference in rates of IPV victimisation according to risk of bias (*Q* = 1.74, *df* = 1, *p* = 1873). Analysis of outliers compared pooled estimates from all studies to pooled estimates when excluding outliers (identified as studies with 95% CIs that did not overlap with the CI for the pooled estimate). Although multiple outliers were identified across analyses, removing these did not substantially alter prevalence estimates, with the largest difference being around 3% (see Appendix A).

## 4. Discussion

This paper provides overall estimates of the prevalence of recent IPV perpetration and victimisation among AD personnel and veterans, when derived from studies using population-based designs and population screening. The search identified *k* = 31 studies that comprised different approaches to measuring IPV, investigations of women and men, general samples of AD personnel and veterans, along with users of specific health services. These reported variable findings regarding IPV prevalence, which are likely to reflect factors including different contexts and approaches to measurement. While the distribution of findings was considered, the primary analyses provide pooled estimates of prevalence based on all available evidence, and thus yield estimates of recent IPV that are most generalisable across contexts. These analyses indicated around 13% of all AD personnel and veterans that reported any recent IPV perpetration, and around 21% that reported any recent victimisation. By way of comparison, analogous figures from civilian studies suggest around 6% of the U.S. population that report any past-year IPV perpetration [70], while around 5% of both men and women report victimisation [71,72]. The current review thus indicates that IPV perpetration and victimisation both occur at elevated levels among AD personnel and veterans, and accordingly comprise major concerns among military and ex-service personnel, as well as their family members.

The review also examined the distribution of estimates across sample characteristics and identified high rates of IPV in studies of both women and men, with no significant differences observed for any perpetration (28.8% for women and 15.8% for men) or victimisation (24.2% and 28.1% for women and men). Findings of gender symmetry in IPV prevalence are not uncommon in population surveys [18], and this highlights important caveats for interpretation of findings of studies in the current review. Most of these used brief screening tools that did not measure impacts of violence (e.g., injury), which are typically greater among women survivors of IPV, when compared to men [72,73]. Furthermore, the current review identified few studies that provided data regarding coercive and controlling behaviours, which are also expected to be perpetrated mainly by men against women [28]. Instead, the findings of this review indicate rates of presumably heterogeneous forms of violence that reflect diverse dynamics and contexts for behaviour. These include coercive and controlling behaviours, as well as many instances of ‘situational’ violence that are expected to be less gendered and perpetrated commonly by both women and men.

Findings of comparable rates of IPV across women and men should be viewed in relation to demographic trends indicating that men still comprise the large majority of AD personnel, and thus veterans, in developed countries such as the U.S. [74]. This suggests that men’s violence will have greatest consequences in absolute terms and at the population-level, and it thus holds that men’s use of violence should comprise a foremost priority for initiatives aiming to reduce the burden of IPV among AD personnel and veterans. However, the results highlight that women AD personnel and veterans also use violence in intimate relationships, and that men’s exposure to IPV are concerns that require attention. This is consistent with emerging literature on women who use force in relationships [75], and nascent evidence regarding men who experience IPV [76]. Such research suggests that these behaviours can reflect unique dynamics and experiences, and thus signal the need for tailored approaches to prevention and response.

Further analyses indicated differences in findings according to serving status, whereby significantly higher rates of any IPV perpetration were typically observed among studies of veterans (31.8%), when compared to AD personnel (5.2%), while there was a modest (and non-significant) difference between rates of victimisation among veterans (24.3%) and AD personnel (13.3%). Findings of elevated IPV perpetration among veterans should be viewed cautiously, as they may reflect reluctance to disclose abuse among current military members. However, genuine differences may be attributed to stressors that characterise the transition out of military service (e.g., loss of identity and security) [77,78], as well as mental health problems among veterans [79]. These include problems with substance use and PTSD that have been implicated in violent behaviours [10,80,81], and may have additional indirect impacts; for example, via difficulties maintaining employment and financial stressors which can increase risk of violence [82]. In contrast, the modest differences observed for victimisation could suggest that while veterans experience additional vulnerability to this non-military trauma, AD personnel are also exposed at high rates and signal important concerns across populations.

Analyses indicated another discernible effect associated with study setting, whereby higher rates of IPV perpetration were typically observed in studies of military or veteran-specific health services (33.7%), when compared to general samples of AD personnel or veterans (9.9%). In contrast, there was a modest (and non-significant) trend towards lower rates of IPV victimisation identified in health services (15.3%), relative to general samples (24.2%). Findings of elevated perpetration in health service contexts should be viewed as provisional given the small number of studies that operationalised this sub-group. However, the results seem plausible in light of mental health problems that are common presenting problems in such settings [79,83], as well as expectations of vigilance among service providers for violent behaviour. In part, this may result from screening for general violence among veterans [84,85], which could support identification of individuals who use violence in their relationships. In contrast, the trend towards lower rates of victimisation in health services could reflect low rates of inquiry by service providers, sub-optimal quality of inquiry (e.g., impersonal, rushed, lack of eye-contact, and sensitivity), or other obstacles such as concerns about having IPV documented in medical records [86,87]. There is broader literature suggesting that such issues can affect identification of IPV in health services [88,89], and has informed interventions that aim to increase IPV disclosure and appropriate responses in such contexts [90,91].

An additional objective of this review was to examine variability across types of IPV, and analyses were conducted to estimate the extent of physical, sexual, or psychological IPV. These indicated around 19.4% of women (14.3% of men) reported recent perpetration of physical IPV, while around 7.4% of women (10.1% of men) reported physical IPV victimisation. These figures can be compared with prior reviews that produced higher point estimates for both past-year physical IPV perpetration (22.0% for women and 27.0% for men) [13] and victimisation (16.2% for women and 21.0% for men) [12]. Such dissimilarities are likely attributed to methodological differences, including the current focus on population-based designs and population screening studies. Analyses regarding other types of IPV should be viewed in relation to smaller numbers of studies of sexual and psychological IPV perpetration, and few studies overall of IPV victimisation among men. However, the available evidence typically indicated that the highest rates were observed for psychological IPV (for example, around 22.0% of women reported recent psychological IPV, compared to 7.4% for physical violence). Rates of sexual IPV were lower but still non-trivial, with around 5.2% of women reporting sexual IPV victimisation in the past year. Sexual IPV is highly stigmatized and likely underreported [1]. As far as we know, there have been no prior attempts to meta-analyse findings regarding sexual or psychological violence among AD personnel or veterans, and the current results provide new evidence of the preponderance of psychological IPV in military and ex-service contexts.

Finally, the review identified small numbers of studies that reported findings regarding impacts and context for IPV, and no studies used comprehensive measures of coercive and controlling behaviours. As such, it identifies further caveats regarding interpretation of evidence regarding IPV in military and veteran populations, which does not typically address violence that is characterized by impact, or that occurs in the context of coercive and controlling behaviours. There is little that can be inferred currently from the small number of informative studies, owing to heterogeneous methods and variable findings. This includes findings of lower rates of IPV with ‘impact’ that were reported by AD air force personnel from the U.S. [40,46], alongside studies which identified much more common reports of stalking experiences by an intimate partner [63], and fear of partner [44]. Accordingly, this review highlights important gaps in understanding of violence impacts and coercive and controlling behaviours and contexts for IPV. This is notwithstanding features of the military cultural context that could plausibly influence tendencies to use coercion and control in relationships, which include hierarchical and male dominated organizational structures, as well as training programs that can reinforce the use of force in certain contexts.

### Limitations

The current findings should be considered in relation to limitations of available evidence. As noted, the studies reported scant data regarding the impact and context for IPV, and the quantitative findings do not necessarily indicate coercive controlling forms of violence. The review also does not indicate whether violence was unidirectional or bidirectional, including whether violence was used in self-defence. Notwithstanding the focus on studies that are defensible for purposes of estimating prevalence, the findings may still underestimate IPV for various reasons. For example, many perpetrators may be reluctant to disclose violence, while partners that use controlling behaviour can prevent victims from participating in surveys. Furthermore, IPV victimisation may also be associated with stigma that may discourage disclosures. There were many studies which only administered IPV measures to participants who were in current relationships, which may also distort estimates by excluding consideration of violent behaviours perpetrated by ex-partners [21]. Only a single study reported data regarding gender identities apart from men and women [53]. Finally, most studies were situated in the U.S. and may have limited generalizability to other jurisdictions.

The current findings should also be viewed in relation to limitations of the review methodology. The initial phase of the search strategy (screening of titles and abstracts) involved only a subset of records that were screened by two reviewers, and it is possible that some potentially eligible studies were missed. There were several studies that reported findings adjusted by survey weights, and the unweighted sample size and number of cases were used to calculate prevalence for these studies. This was because weighted prevalence figures are difficult to incorporate into conventional meta-analysis programs, such as the *metaprop* function, which calculates prevalence on the basis of number of events and sample size, which are manipulated following application of sampling weights. The review considered studies that reported findings regarding recent IPV, which included past year, past six months, past month, and any time in the current relationship. This reflects a more inclusive approach when compared with reviews that focused on past year IPV, for example. This was intended to minimise bias associated with omitting potentially informative studies, but also introduces heterogeneity into the findings, with studies of IPV across shorter intervals likely producing lower estimates when compared with studies of experiences in the past year or current relationship. Conversely, the sensitivity analyses suggested that studies of IPV perpetration which were characterized by risk of bias typically produced higher rates when compared with studies having low risk of bias, and this may suggest overestimates of relevant figures.

Population-based studies were excluded if they were based on small samples (*n* < 100), while population screening studies were excluded if response rates were low (<60%). These exclusions were intended to address specific sources of bias in studies, but they could also have introduced bias associated with omission of informative data. The focus on IPV reported by AD personnel and veterans excluded studies that addressed IPV reported by family members, as well as agency data (e.g., police records). Furthermore, the review excluded studies of general types of family violence, including child exposure. It considered reports published in peer-reviewed journals and did not capture studies reported only in the grey literature. Some sub-group analyses were constrained by small numbers of studies, and comparisons may be affected by low power. Finally, there were high levels of between-study heterogeneity in estimates that was not explained by the sub-group analyses. Accordingly, the pooled estimates from meta-analyses should be viewed as average figures that are most generalisable across contexts. However, these should be contextualised by consideration of variation across studies that was not explained in this review.

## 5. Conclusions

The current findings indicate that IPV is common among AD personnel and veterans, and thus they signal the need for recognition and responses across military and ex-service settings. This includes responses to IPV victimisation, which while not necessarily attributed to aspects of military service, are still likely to interact with military trauma and account for adverse mental health and psychosocial outcomes. In addition, the review signals the need for responses to IPV perpetration among AD personnel and veterans, which are more likely influenced by the military context, including routine exposure to trauma, and training that can reinforce the use of force for instrumental purposes. These expectations of adverse impacts of IPV, and plausible roles of military experiences in perpetration of violence, provide a strong rationale for relevant agencies and institutions, including government departments, health services, and veteran support organisations, to enhance strategies for addressing IPV among AD and former military members.

The findings highlight the need for investment in resources to meet the needs of IPV victims, which may comprise many AD personnel and veterans (including increasing numbers of women), as well as their current or former partners and children. These may include initiatives which are situated in military and veteran-specific health services and involve training for service providers in signs and first-line responses to IPV, and referral pathways to services for victims [16]. The latter may comprise advocacy programs that can help address immediate safety needs; for example, via guidance on safety planning and provision of counselling and advocacy support for victims on accessing community-based resources (e.g., legal and financial advice services, emergency housing, or shelter access) [92]. Such specialist services could be embedded within military or ex-service agencies, or alternatively, in community-based organisations that have enhanced military cultural competence. Service organizations should routinely inquire about military service so that additional tailored resources and referrals can be considered.

The review also highlights the need for initiatives that focus on addressing use of violence by AD personnel and veterans. Recent literature has considered the accuracy and acceptability of standardized approaches to questioning about IPV use among military personnel [93,94], which could inform identification strategies in health services, while there are also emerging programs for men who use violence that have been implemented in U.S. settings [95,96]. However, this literature is in its infancy and there is a need for additional guidance regarding effective responses to IPV perpetrators. In this context, there should be major investments in IPV prevention strategies, and responses which can ensure that AD personnel and veterans who use violence and wish to change their behaviour are supported to do so. It is critical that these are developed in parallel with broader reforms that prioritise the safety of victims, while holding perpetrators accountable for their behaviours. Structural reforms may include prescribing military and veteran-specific agencies to have responsibilities for addressing IPV, along with enhanced systems to ensure that agencies that have contact with AD personnel and veterans can share information to help assess and manage risk and intervene early to enhance safety.

Finally, the review signals priorities for future research regarding IPV among AD personnel and veterans. Most notably, the results highlight a need for studies outside the U.S., and across jurisdictions including Canada, Australia, and the UK. These should include general samples of AD personnel and veterans, that capture the gendered experiences of women, men, transgender, and non-binary individuals, as well as research in specific health settings. The latter may be marked by differences across jurisdictions in the organization of services delivered to veterans (for example, the UK does not have an extended history of specific services for veterans, with care instead delivered mainly in the publicly funded National Health Service) [97] and family members (who may receive care in separate service systems to veterans in some countries), which could have implications for IPV disclosure. In addition to questions regarding violence impacts, future studies should include measures of coercive and controlling behaviours [26], which should be administered with reference to both current and former intimate relationships. Finally, there is also a need for research to supplement findings of prevalence studies. These include qualitative investigations to increase understanding of the complexity of IPV among AD personnel and veterans, as well as their preferences for intervention, and further studies of identification strategies and interventions in military and veteran-specific settings.

## Figures and Tables

**Figure 1 ijerph-19-08853-f001:**
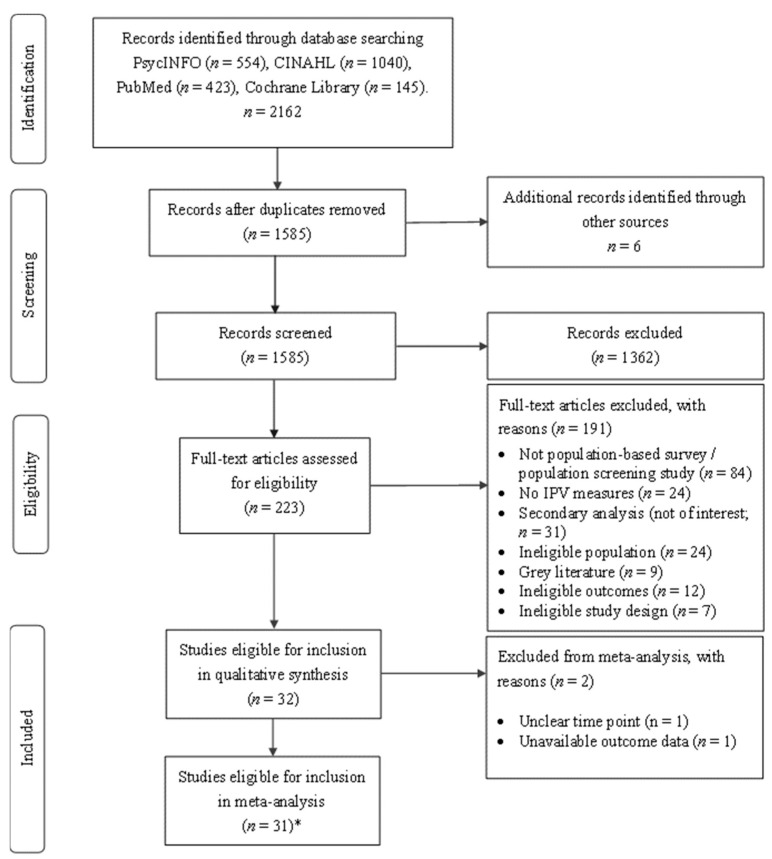
PRISMA flow diagram. Note: * = One paper included two studies [40].

**Table 2 ijerph-19-08853-t002:** Meta-analysis findings reported by gender and IPV type.

	Women Only Samples			Men Only Samples		
	*k*	Estimate (95% CI)	I^2^	*k*	Estimate (95% CI)	I^2^
**Recent IPV Perpetration**						
Any IPV	5	28.8% (14.7–48.7%)	99.4%	7	15.8% (11.8–20.9%)	99.4%
Physical IPV	4	19.4% (13.2–27.8%)	97.3%	6	14.3% (9.6–20.8%)	99.7%
Sexual IPV	1	11.9% (-)	-	1	4.0% (-)	-
Psychological IPV	2	39.5% (11.0–77.6%)	98.1%	1	19.4% (-)	-
**Recent IPV Victimisation**						
Any IPV	9	24.2% (14.4–37.7%)	99.7%	4	28.1% (13.6–49.3%)	99.4%
Physical IPV	6	7.4% (3.5–13.7%)	99.1%	3	10.1% (5.4–18.3%)	94.4%
Sexual IPV	5	5.2% (2.0–13.0%)	98.2%	2	0.8% (0.0–8.8%)	82.0%
Psychological IPV	7	22.0% (10.5–40.4%)	99.7%	4	20.2% (6.8–46.6%)	99.6%

Note. Seven studies provided separate estimates for men and women samples. Weighted means are reported with 95% confidence intervals. Single estimates comprise prevalence estimates as reported by study authors.

**Table 3 ijerph-19-08853-t003:** Subgroup analyses for studies of any recent IPV perpetration and victimisation.

Moderators	*k*	Estimate (95% CI)	*k*	Estimate (95% CI)
	Perpetration		Victimisation	
Gender				
Men	7	15.8% (11.8–20.9%)	4	28.1% (13.6–49.3%)
Women	5	28.8% (14.7–48.7%)	9	24.2% (14.4–37.7%)
Serving status				
Active duty	6	5.2% (1.5–16.9%)	5	13.3% (7.0–23.6%)
Veteran	5	31.8% (16.3–52.6%) *	8	24.3% (13.2–40.1%)
Era of service				
Pre 2001	2	13.6% (5.6–29.4%)	0	-
Post 2001	7	9.1% (2.2–30.6%)	7	23.7% (10.5–45.1%)
Study setting				
General military	9	9.9% (3.3–26.4%)	9	24.2% (12.6–41.2%)
Health service	2	33.7% (23.8–45.2%) *	5	15.3% (10.5–21.7%)
Country				
US	10	12.4% (4.3–30.5%)	12	20.2% (11.8–32.4%)
Non-US	1	16.5% (14.8–18.3%)	2	23.6% (20.8–26.5%)

Note. * = *p* < 0.05.

## Data Availability

Not applicable.

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
