# Peer review of "Intimate Partner Violence (IPV) in Military and Veteran Populations: A Systematic Review of Population-Based Surveys and Population Screening Studies"

_ijerph, 2022, doi:10.3390/ijerph19148853_

Round 1
Reviewer 1 Report
Please see attached word document.

Author Response
Reviewer: 1
In this interesting systematic review and meta-analysis, the authors sought to provide a better estimate of the prevalence of IPV perpetration and victimization among active duty military and veteran populations. By limiting their review to studies that utilized population-based design or population screening strategies, they aimed to find more accurate and representative estimates of IPV prevalence in this population compared to prior studies and reviews, which utilized a more heterogeneous variety of sampling strategies and yielded highly variable estimates. Furthermore, they provide more granular data by examining prevalence of different types of IPV and by conducting sub-group analyses based on gender, service status, era or service, country, and study setting. This systematic review and meta-analysis reveals a high prevalence of both IPV perpetration and victimization in the population of interest, although there was a high degree of heterogeneity among the studies included in the review. One novel aspect of this review, and, in my opinion, one of the most interesting, and one of this article’s greatest strengths, is the focus on IPV context and impact as well as controlling and coercive behaviors. Even though studies addressing IPV context and impact were included in the meta-analysis due to the heterogeneity of IPV context and impact indicators, the data are presented clearly and exhaustively in narrative form, and the authors thus draw our attention to an important gap in the evidence.
The authors provide sufficient background information and convincingly lay out the case for this review’s necessity and novelty. In particular, the methodological issues with prior studies and reviews on this topic are explained in detail, which makes clear the rationale for this review. The aims are stated clearly and flow logically from the background information provided and from the gap in the evidence identified in the introduction.
The statistical methods are sound, and the rationale for not including findings on IPV impact or context in the meta-analysis is appropriate. The tool used for risk of bias assessment is appropriate.
The results are presented plainly and clearly, with an appropriate level of detail. The tables are well-organized and easy to interpret.
In the discussion, the authors summarize the findings concisely, offer cogent hypotheses to explain both expected and unexpected findings, and address limitations of the study extensively.
The conclusion artfully contextualizes the importance of the findings, describes their implications, and suggests intelligent measures to address the problem of the high prevalence of IPV in military and veteran populations.
However, there are a few major and minor methodological issues that should be addressed.
Major issue:
- Only one reviewer conducted the title and abstract screening and the data extraction. This is especially problematic for the title and abstract screening, since it seems as though not all articles were even screened by the same reviewer. This potentially introduces bias as there is no way to check inter-rater reliability, and could result in missing a substantially higher proportion of relevant articles.1,2 The quality of this systematic review would be substantially improved by having at least one other reviewer for both the title and abstract screening and data extraction. Discrepancies between the two reviewers could be resolved by discussion or through adjudication by a third reviewer, as was done in the full text review phase.
Unfortunately, we did not have the resources to have multiple reviewers screen all titles and abstracts. However, to assess and enhance consistency in this stage of screening, there was a pilot phase in which the two reviewers independently screened around 20% of the same records. This pilot phase identified disagreements for <5% of records, and these were resolved through discussion involving a third reviewer, with a view towards improving consistency for the remainder of the screening process (where the two reviewers each screened 40% of records). This detail had been omitted from the paper in order to reduce length, but has now been inserted (p. 4, line 192-196). We have also made an insertion to highlight this as a limitation of the review methodology (p. 21, line 663-665).
Minor issues/clarifications:
- The rationale for excluding small sample and low response rate studies (page 4, section 2.2, Eligibility Criteria) makes sense, however this exclusion itself could introduce bias. An alternative approach would be to include these studies but consider small sample size and low response rates in the risk of bias assessment (as was done for studies involving probability‐based sampling) and/or conduct a sensitivity analysis comparing the original analysis to a new analysis with these studies included.
We agree there would be alternative approaches to managing small samples and low response rates, and these would all be associated with strengths and weaknesses. Given the broad scope of this review, which includes a relatively large number of analyses which are focused on both IPV victimisation and perpetration, we are reluctant to add further complexity of the paper via additional risk of bias assessments and sensitivity analyses. Given that we believe our approach is defensible (although subject to limitations), we have maintained this in the revised version of the paper. However, we have made an insertion in the limitations to acknowledge these and alternative approaches (p. 21-22, ling 681-684).
- Please specify which author(s) conducted the risk of bias assessment
Two authors conducted risk of bias assessments and this information has been included in the revised paper (p. 4, line 203).
- Did a library scientist with experience in systematic review participate in devising the search strategy?
The review team for this project had extensive experience in the design and implementation of systematic search strategies. This includes the first author (SC) and third author (DK), who have collectively published more than a dozen systematic and scoping reviews (including Cochrane standard systematic reviews) and were responsible for designing the search strategy. Accordingly, we did not have an identified need for consultation with an external library scientist. We have not made any changes in response to this comment.
- Did any of the studies included in the review specify whether participants were in heterosexual or same-sex relationships?
We did not systematically extract information regarding whether studies expressly reported participants were in heterosexual relationships. However, there were no studies identified that reported participants were in same-sex relationships, and we presume that this number was very low.
- What was the rationale for choosing 2001 as the cut-off year for era of service?
This year corresponds to the 9/11 terrorist attacks in the U.S., which were catalysts for a distinctive era of ‘contemporary’ military engagements in Afghanistan and Iraq.
- In table S3, under perpetration, the articles classified according to sampling strategy do not add up to 12. Was there another sampling strategy that is not shown in the table? If so, please add.
We are grateful to the reviewer for identifying this. The figure was erroneous and has now been corrected.
- On page 16, section 3.3.1 (Recent IPV Perpetration) it is stated that k=11 studies were included in the meta-analysis (out of a total of 12 studies which addressed IPV perpetration), yet according to table S3, all 12 studies on IPV perpetration were included in the meta-analysis. Similarly, on page 17, section 3.3.2 (Recent IPV Victimisation) it is stated that k=14 studies were included in the meta-analysis, yet according to table S3, all 25 studies addressing IPV victimization were included. Please clarify, and specify which studies were included in the meta-analysis (citation is sufficient).
These discrepancies are due to our initial search strategy and analyses including studies that reported estimates of both lifetime and recent IPV, while the meta-analyses in this paper focussed on just recent IPV (analyses of lifetime IPV were not reported to reduce length). We have made a number of minor corrections in order to resolve these inconsistencies:
- We have removed the text 'included in the MA' from the title of Table S3. This was incorrect as this table also includes studies that reported lifetime estimates and were included in the review, but not the meta-analyses.
- We have corrected the text in the manuscript (p. 5, line 233-234) to clarify that studies which reported lifetime estimates of IPV were included among eligible studies, while only those that reported recent estimates are included in the analyses;
- We have used superscript letters (a, b) in Table 1 to indicate studies that were not included in the analyses (and explained this in the table footnote).
- We have included in Table S3 information about the number of studies that reported lifetime versus recent IPV estimates.
Overall, this is a novel, interesting, and very well-written review which certainly has merit and would constitute an important addition to the evidence on IPV prevalence in military and veteran populations, as long as the methodological issues identified above are addressed.
We greatly appreciate the time taken to review this paper.
Reviewer 2 Report
This is a methodologically sound, thoroughly researched manuscript. The presentation is clear and the results are interesting. I suggest minor revisions:
Spelling- Victimization / Victimisation – Please double check journal requirements regarding US vs European
Please add some initial clarity to set the stage up front– systematic review of US personnel? Worldwide? Just a sentence or two.
The introduction is easy to follow and establishes the gap in the literature. The specific objectives, however, should delineate the parameters for clarity, e.g., adults, international or US and why, etc. Or only specific countries given the studies at hand? Lines 131-140
Lines 141-144 – needs a brief description of PRISMA for anyone who may not be familiar. Why this approach over another?
The search strategy and exclusion criteria are robust and well laid out.
Line 220 “data were” – change data to plural throughout.
Line 273, briefly explain the conflict scale; lines 274-279 – what are the main differences among these scales? Line 316-same (In fact, a brief mention at the beginning of the methods on types of scales and their differences/similarities would be helpful).
The choice to summarize some of the data, such as withholding birth control, was smart. The information is still meaningful and should stay in the paper.
The results are very thorough, and the subheadings are key to following the very detailed paper. The discussion reads as a recap of the results or as if introducing new results. I suggest subheadings in this section that align with the results. Move any results from the discussion section back to the results section and write it with interest, i.e., the “so what.” At present, in its dry and rigid form, it doesn’t help the reader understand the importance of the results. Much of the conclusion could be moved to the discussion. Parts of the conclusion read more like a discussion. Button up the conclusion to focus on the most important findings. What do these findings imply for public health? Military leaders?
Depending on journal requirements, I suggest moving the limitations, which are well thought out, to the end of the manuscript.
Author Response
Reviewer: 2
This is a methodologically sound, thoroughly researched manuscript. The presentation is clear and the results are interesting. I suggest minor revisions:
Spelling- Victimization / Victimisation – Please double check journal requirements regarding US vs European
We have checked journal requirements, and according to the journal publisher, American English and UK English are both suitable assuming consistency throughout the manuscript. We have reviewed the manuscript to ensure consistency in UK English throughout.
Please add some initial clarity to set the stage up front– systematic review of US personnel? Worldwide? Just a sentence or two.
The review considered studies from any international jurisdiction (not just the U.S.), and we have made multiple insertions to clarify this at the end of the introduction (p. 3, line 137, line 143), and at the beginning of the method (p. 3, line 157-158).
The introduction is easy to follow and establishes the gap in the literature. The specific objectives, however, should delineate the parameters for clarity, e.g., adults, international or US and why, etc. Or only specific countries given the studies at hand? Lines 131-140.
As noted above, we have made insertions at the end of the introduction to indicate that this review encompassed studies from any international jurisdiction. The other key parameters for the review (e.g., focus on population-based designs) are already clearly identified.
Lines 141-144 – needs a brief description of PRISMA for anyone who may not be familiar. Why this approach over another?
We have made an insertion at the beginning of the method to introduce the PRISMA guidelines (p.3, line 153-154). However, these guidelines are used extensively in the systematic review literature, and we are unaware of alternative approaches that would warrant justification for using PRISMA as a preferred alternative.
The search strategy and exclusion criteria are robust and well laid out.
No response required.
Line 220 “data were” – change data to plural throughout.
This has been corrected throughout the manuscript.
Line 273, briefly explain the conflict scale; lines 274-279 – what are the main differences among these scales? Line 316-same (In fact, a brief mention at the beginning of the methods on types of scales and their differences/similarities would be helpful).
Given the range of measures of IPV perpetration and victimisation that have been considered across studies, a meaningful description of validated scales, including their similarities and differences, would add substantial length to the paper. Given that we have already provided a particularly in-depth account of measures of IPV impact and context, we believe that adding further length which relates to validated and widely recognised scales is out of scope. We have not made changes in response to this comment.
The choice to summarize some of the data, such as withholding birth control, was smart. The information is still meaningful and should stay in the paper.
No response required.
The results are very thorough, and the subheadings are key to following the very detailed paper. The discussion reads as a recap of the results or as if introducing new results. I suggest subheadings in this section that align with the results. Move any results from the discussion section back to the results section and write it with interest, i.e., the “so what.” At present, in its dry and rigid form, it doesn’t help the reader understand the importance of the results. Much of the conclusion could be moved to the discussion. Parts of the conclusion read more like a discussion. Button up the conclusion to focus on the most important findings. What do these findings imply for public health? Military leaders?
We appreciate the suggested alternative approach to structuring the discussion. However, we are reluctant to make substantial changes to the organisation of this section, given that the discussion seems to have been positively received by all the other reviewers (Reviewer 1, for example, described the discussion and conclusion as cogent and artful). We recognise that readers will have different preferences for how material in a discussion section might be organised, but we believe our approach is distinct from the results (there are no new results presented in the discussion, although we do revisit key findings as they are discussed in the context of relevant literature), is logically organised, and includes clear statements of the significance of the findings. Accordingly, we have not made changes to the discussion in response to this comment.
Depending on journal requirements, I suggest moving the limitations, which are well thought out, to the end of the manuscript.
Given that we have maintained our original approach to organising the discussion, we believe the limitations are appropriately situated in the context of the discussion, and have not made changes to the organisation of the paper.
Reviewer 3 Report
Overall, this manuscript is systematically well-written, and the findings are impactful, considering that studies on IPV among military and veteran populations are scant. The introduction section could be improved with the addition of existing study findings about the comparison of IPV characteristics between Active Duty (AD) military personnel and veterans.
Author Response
Reviewer: 3
Overall, this manuscript is systematically well-written, and the findings are impactful, considering that studies on IPV among military and veteran populations are scant. The introduction section could be improved with the addition of existing study findings about the comparison of IPV characteristics between Active Duty (AD) military personnel and veterans.
We have made an insertion in the introduction regarding the comparison between AD personnel and veterans (p. 2, line 63-64). This literature is then addressed more expansively in the discussion section (p. 20, line 576-590).
Reviewer 4 Report
The paper is focused on a very interesting subject. This is well structured and well argued. The article can be published without any changes.
Author Response
Reviewer: 4
The paper is focused on a very interesting subject. This is well structured and well argued. The article can be published without any changes.
We appreciate the feedback and time taken to review this manuscript.
Round 2
Reviewer 1 Report
Thank you for thoroughly addressing the issues pointed out. I believe that clarifying the methodology used for screening articles, explaining its rationale, and describing its potential limitations more clearly, as you appropriately did, greatly improves the quality of the manuscript, and I am recommending it for publication in its current form.